

# Plantar load distribution with centers of gravity balance and rearfoot posture in daily lives of Taiwanese college elite table tennis players: a cross-sectional study

Tong-Hsien Chow[1],[*] and Yu-Ling Lee[2],[3],[*]

[1] Department of Sports Science, R.O.C. Military Academy, Kaohsiung, Taiwan
[2] Department of Tourism Leisure and Health, Deh Yu College of Nursing and Health, Keelung, Taiwan
[3] Department of Physical Education and Sport Sciences, National Taiwan Normal University, Taipei, Taiwan
[*] These authors contributed equally to this work.

Corresponding author
Tong-Hsien Chow,
thchowma@gmail.com

## ABSTRACT

**Background:** Table tennis is an asymmetric sport involving the powerful forward swing of the upper limbs depends on the solid support of the lower limbs. The foot drive really affects the weight balance and stroke accuracy even though the distance and momentum of the lower limb displacement are limited within a limited range. Given that previous research on table tennis has typically focused on the footwork and stroke performance of professional players, the study aimed to investigate the daily static and dynamic plantar load distribution as well as the centers of gravity balance and rearfoot posture among Taiwanese college elite table tennis players.

**Methods:** This is a cross-sectional study of 70 elite male table tennis players (age: 20.0 ± 0.9 years; height: 173.4 ± 5.1 cm, weight: 67.6 ± 5.3 kg, experience: 10.0 ± 1.6 years) and 77 amateur table tennis players of the same gender (age: 20.1 ± 0.8 years, height: 167.4 ± 4.4 cm, weight: 64.3 ± 4.0 kg, experience: 4.4 ± 1.2 years) from Taiwanese universities. The JC Mat optical plantar pressure analyzer was applied to determine the plantar load distribution along with arch index (AI) and centers of gravity balance. Assessment of rearfoot postural alignment was mainly used to contrast the performance of the centers of gravity balance.

**Results:** The static arch indices of both feet in the elite group were symmetrical and considered normal arches (AI: 0.22 ± 0.07) during their non-training and non-competition daily lives. Their static plantar loads were symmetrically concentrated on the bipedal lateral metatarsals ($P < 0.05$) as well as shifted to the medial and lateral heels ($P < 0.05$) and the lateral metatarsals ($P < 0.05$) during the walking midstance phase. Additionally, the plantar loads were mainly applied to the bipedal medial ($P < 0.01$) and lateral heels ($P < 0.05$) during the transitional changes between both states. Elite athletes had symmetrical and evenly distributed centers of gravity on both feet (left: 50.03 ± 4.47%; right: 49.97 ± 4.47%) when standing statically, along with symmetrical rearfoot angles and neutral position of the subtalar joint (left: 2.73 ± 2.30°; right: 2.70 ± 2.32°) even though they were statistically lower than those of the amateur athletes ($P < 0.05$).

**Conclusions:** The daily static and dynamic foot patterns of Taiwanese college elite table tennis players were characterized by plantar load distribution on the lateral

metatarsals and the entire calcaneus along with balanced centers of gravity and normal rearfoot posture. This foot and posture layout outlines the excellent athletic performance of the foot and ankle in professional athletes. Portions of this text were previously published as part of a preprint (https://doi.org/10.21203/rs.3.rs-2993403/v1).

# INTRODUCTION

Table tennis is an asymmetrical sport discipline that requires simultaneous coordination of the upper limb handwork with lower limb footwork and trunk rotation to complete stroke performance instantaneously (*He et al., 2022*). Table tennis players usually complete a series of complex spatial movements such as acceleration and deceleration along with rapid movement and direction change as well as body balance control under the condition of body coordination, and the driving of these movements will help to exhibit optimal stroke performance (*Yu et al., 2019*). The powerful forward swing of the upper limbs requires the agile transposition of the lower limbs, which can be regarded as the origin of the kinetic chain in table tennis (*He et al., 2021*). Yet, the support of the feet on the ground and the footwork affect the balance of the body and the accuracy of strokes (*Yu et al., 2019*), consequently, agile footwork is considered to be an important physical quality that reflects the technical skill level of athletes (*Bańkosz & Winiarski, 2018*). Optimizing plantar and ankle movement strategies and the flexibility of the ankle muscle groups are thought to contribute to energy transfer in the kinetic chain. The stability of foot and ankle support upon landing has also critically received attention in the field of table tennis coaching research (*He et al., 2022*).

Previous studies mentioned that professional table tennis athletes have agile footwork due to their unique processing skills in foot motion control in the study of the center of pressure (COP) trajectory during topspin forehand loop performance (*Fu et al., 2016*). There are significant differences between professional athletes and beginners in the contribution of the lower limbs in table tennis chasse footwork (*Yu et al., 2019*). Professional athletes have greater foot control and better technical stability, which results in a shorter chasse footwork time and more forward swing time compared to beginners, suggesting that professional athletes have a stable center of gravity shift for the chasse footwork and a superior ability to control body balance (*Yu et al., 2019*). Plantar pressure assessment is considered useful in examining plantar load performance in athletes of different levels in competition (*Wong, Lee & Lam, 2020*). The study by *Yu et al. (2019)* showed that professional table tennis athletes had higher relative plantar loads on the other toes and the lateral forefoot, while lower on the medial part of the forefoot and rearfoot during one entire motion cycle compared to beginners. Significant increase in force-time integral (FTI) on the rearfoot region while serving with a standing position (*Yu et al., 2018*). During typical footwork, however, the bipedal peak pressures and relative plantar

loads are mainly concentrated on the lateral forefoot and medial-lateral rearfoot, while the force on the midfoot is less obvious (*Li, 2022*). Additionally, during the support phase of the striding gait, the main bearing region of the athlete's plantar pressure is the forefoot, and the peak pressure on the forefoot can reach about 50% of the body weight (*Bańkosz & Winiarski, 2020*). Similar results were argued in the study by *He et al. (2022)* that the athlete's fifth metatarsal, lateral forefoot, medial and lateral rearfoot absorb impact-transmitted forces during landing, resulting in massive activation of the muscles around the ankle and subtalar joints to maintain foot stability during the landing phase.

Agile gait behavior and stable center of gravity control are both considered to be closely related to the rearfoot posture of athletes, especially for professional players (*Li, 2022*). According to the research of *Iordan et al. (2011)*, professional table tennis athletes typically undergo special sports training around the age of 6–8 years old; thus, a repetitive rapid unilateral movement can eventually contribute to postural deficits. Furthermore, professional athletes have evolved significant rearfoot pronation with greater forefoot abduction in backward-end footwork to facilitate body postural balance during the chasse movement (*Yu et al., 2019*). Conversely, changes in body posture and gait behavior of table tennis players are mainly caused by twisting at the trunk level and various specific movements performed by the musculoskeletal system (*Wong, Lee & Lam, 2020*).

Since previous studies involving table tennis have primarily focused on specific footwork and the impact of specific serving and intercepting actions on the athletes' plantar load distribution. Yet, these studies rarely discuss the performance of static and dynamic plantar load distribution experienced by table tennis players in their daily lives, and little has been reported about the changes in plantar load that amateur athletes may experience when they progress to become elite athletes through repeated training and competition. Given the above background, this study inherits the context and methodologies of previous studies on examining the relationship between static and dynamic foot pressure profiles with lower limb pain profiles in different elite athletes (*Chow & Hsu, 2022*; *Chow et al., 2022*; *Chow, Chen & Hsu, 2021*; *Chow et al., 2021*; *Chow, Chen & Wang, 2018*). Based on the overall performance of the correlation among arch index (AI), plantar load distribution, centers of gravity balance and rearfoot postural alignment may be associated with the footwork experiences and interception skills of college elite table tennis players after long-term training. Therefore, the study aimed to explore the characteristics of plantar load distribution as well as the centers of gravity balance and rearfoot posture of elite table tennis players in their daily static stance and natural gait during non-training and non-competition periods. It was hypothesized that the elite table tennis players in static stance belonged to the normal-arched foot and that their plantar loads were exerted more on the forefoot while being transferred to the lateral forefoot and entire rearfoot during the walking midstance phase. The performance of the subtalar neutral position of the rearfoot was correlated with their well-trained superior interception technique along with balanced control ability and agile gait, that is, their rearfoot postures were normal and the centers of gravity were stable and evenly distributed over their feet.

# MATERIALS AND METHODS

## Participants

A cross-sectional study was conducted on Taiwanese college-level elite table tennis players during their non-training and non-competition periods. The study recruited 147 eligible male college and university undergraduates over a 2-year period and divided them into two groups: 70 elite table tennis players (referred to as the elite group) and 77 amateur table tennis players (referred to as the amateur group). Participants in both groups came from a relatively homogeneous population who had similar ages, body types and learning styles, and the only difference lies in their table tennis training intensity and regular workout schedules as well as their experience at different levels of competition. Participants in the elite group of the present study were identified as the division A of table tennis players with a right-handed preference and had at least eight consecutive years of table tennis training and competition experiences in the National University and College Athletic Games, the National Table Tennis Championship and The Selective Trial of Table Tennis National Representatives in Taiwan. They were selected from 92 male collegiate division A table tennis players who were recruited from the National Taiwan Normal University, National Taiwan University of Sport, University of Taipei, National United University, National University of Tainan, National Taitung University, National Formosa University, National Taipei University of Technology, National Kaohsiung University of Science and Technology, National Pingtung University of Science and Technology, Fu Jen Catholic University, Providence University, Chinese Culture University, Feng Chia University, Shih Chien University and St. John's University. In the process of recruiting eligible elite table tennis players, there was about a 24% drop-out rate mainly due to the following reasons: (1) absence rate from experiments; (2) self-disclosure of previous fractures or surgeries and presentation of hospital certificates; (3) dislocations or ligament tears in the lower extremities within the last 6 months; (4) have other sports expertise or have received professional sports training other than table tennis; (5) the body mass index (BMI) outside the healthy physical range of 18.5 to 23.9 established by the World Health Organization (WHO) and Asia-Pacific guidelines and (6) records of individual-related competition or training injuries such as calcaneal spurs, skeletal arthritis and lower limb neuropathy provided by their school's coaches or athletic trainers. According to the survey, the weekly schedule of specific interval training for these elite athletes includes aerobic and anaerobic endurance training. Their regular table tennis tactical training courses include multi-ball drills and rally drill. For the amateur group, there were 77 participants selected from 95 male collegiate recreational table tennis players without any sports expertise and professional training who met the above conditions. All experiments in this study were conducted in accordance with the guidelines set by the National Taiwan University Ethics Committee in Taipei, Taiwan, following a rigorous evaluation of the investigators' ethics approval application on 13 June 2015 (NTU-REC No.: 201506ES016) in accordance with the Declaration of Helsinki recommendations. All participants agreed and signed written informed consent form after being informed by the researcher and fully understanding the purpose and content, experimental procedures and requirements of this study.

## Instruments

The study inherits the repeatability and reproducibility of previous studies using the JC Mat optical plantar pressure analyzer (View Grand International Co., Ltd, New Taipei City, Taiwan; sampling frequency of 15 Hz) (*Chow & Hsu, 2022*; *Chow et al., 2022*; *Chow, Chen & Hsu, 2021*; *Chow et al., 2021*; *Chow, Chen & Wang, 2018*; *Wang, Dommati & Cheng, 2019*). The repeatability and reproducibility of the measurement were based on the fact that before conducting the plantar pressure measurement, several weights of proportional weight were used to calibrate the pressure linearity test on the instrument. In this process, it was possible to determine the grayscale diagram of the linear relationship between weight and pressure and the color scale diagram of the linear relationship between weight and pressure within the range of the JC Mat sensing pad, as well as the scale bar used for weight-area correction (data not shown). In addition, the instrument is equipped with FPDS-Pro software that enables researchers to synchronize the measurement of certain parameters from color footprints and real barefoot images of subjects, such as arch index (AI), plantar load distribution, centers of gravity balance and toe angle. Participants' static and dynamic plantar load distributions and centers of gravity balance measurement were the main exploratory factors in this study.

## Procedures

Before the experiment, the physiological characteristics of the participants were surveyed and recorded in detail. The experiments were conducted in the classrooms of each participant's school. In order to ensure the consistency and reliability of the research, the experimental time was scheduled in the morning. In both static and dynamic plantar load measurements, each participant was instructed step-by-step by the same researcher to perform the measurement procedure of the JC Mat optical plantar pressure analyzer described in previous studies to obtain the preliminary results of the distributions in plantar load and centers of gravity (*Chow & Hsu, 2022*; *Chow et al., 2022*; *Chow, Chen & Hsu, 2021*; *Chow et al., 2021*; *Chow, Chen & Wang, 2018*). During the measurement of dynamic plantar load, each participant was first guided to experience walking barefoot back and forth at their own comfortable and stable pace on a two-meter long walkway (200 cm × 70 cm) with a built-in JC Mat. After three rounds of walking back and forth to familiarize themselves with the pre-test process, each participant could officially enter the experiment. During the testing process, each participant was asked to perform three rounds of back-and-forth walking trials on the walkway at their own daily comfortable pace until the researcher was able to accurately record the dynamic plantar pressure of each foot of the participant at least three times, that is, each foot of the participant completely stepped on the sensing pad of the JC Mat that marks the sensor range and measurement area. At the same time, the researcher could instantly observe and collect preliminary data about dynamic plantar pressure, centers of gravity balance and traveling lines from each foot *via* the built-in FPDS-Pro program of the JC Mat external computer. After the experiment, the researcher was able to use slow-motion videos at 1/2 to 1/16-fold speed to analyze and identify the PPDs of the footprints that are completely in contact with the ground during the midstance phase of walking (*i.e.*, the forefoot, midfoot and rearfoot

all appear completely within the captured footprint of the single-leg fully supported stance, which is archived at the time point when the body's centers of gravity are centered on the supporting leg.). Each foot of the participant has analyzed three times and the average of the three results was used to determine dynamic plantar load distribution. An assessment of rearfoot postural alignment was conducted immediately in the same space following the completion of the above study procedures.

## Foot data analysis

Each participant has completed each research procedure throughout the entire research process, and those with incomplete original data were excluded from statistical analyses. After the entire experiment was completed, a detailed analysis of foot parameters such as AI, plantar load distribution and centers of gravity balance of each participant was performed by the same researcher through the built-in FPDS-Pro program on the JC Mat. The percentage of relative plantar load is a variable used to evaluate plantar load distribution. This variable presents the load-bearing effect of the weight per square centimeter in a specific region of the plantar. The AI is a variable defined as the ratio of the area of the middle third of the footprint (midfoot) divided by the area of the entire footprint excluding the toes (*Chow & Hsu, 2022*; *Chow et al., 2022*; *Chow, Chen & Hsu, 2021*; *Chow et al., 2021*; *Chow, Chen & Wang, 2018*; *Cavanagh & Rodgers, 1987*). Further, the centers of gravity balance is a variable that considers the difference in the total percentage of relative plantar load between the participant's left and right feet.

In addition, the static rearfoot angle was calculated using the Biomech 2019-posture analysis software (Loran Engineering SrL, EmiliaRomagna, Italy; sampling frequency of 100 Hz). In terms of analyzing plantar load distributions, the FPDS-Pro program was used to equally divide the captured footprint image (excluding the toes) into six subregions, each of which corresponds to the relative position of the specific anatomical structure of the foot. They were defined in order from the anterolateral to the posteromedial aspects of the foot as lateral metatarsal (LM), lateral longitudinal arch (LLA), lateral heel (LH), medial metatarsal (MM), medial longitudinal arch (MLA) and medial heel (MH), respectively (*Chow & Hsu, 2022*; *Chow, Chen & Wang, 2018*). Additionally, these six subregions could be further merged with each other into the five regions to facilitate direct observation of the large-scale plantar load distributions, centers of gravity distributions and balance performance of participants' feet. Likewise, these five regions were defined and combined sequentially from the front to the back and from the lateral to the medial aspects of the foot as forefoot (LM and MM), midfoot (LLA and MLA), rearfoot (LH and MH), lateral foot (LM, LLA and LH) and medial foot (MM, MLA and MH), respectively (*Chow & Hsu, 2022*). By analyzing the results of static and dynamic plantar pressure measurements, we were able to collect and determine the load distribution in specific regions/subregions and centers of gravity balance of participants' feet under the two states, and further evaluate the transitional changes between the two states. The methods and detailed procedures for the static and dynamic plantar pressure and rearfoot postural alignment were conducted as described in previous studies (*Chow & Hsu, 2022*; *Chow et al., 2022*; *Chow, Chen & Hsu, 2021*; *Chow et al., 2021*;

**Table 1 Basic demographic characteristics of the participants.**

| Characteristic | Amateur group ($n$ = 77) | Elite group ($n$ = 70) |
|---|---|---|
| Age (years) | 20.1 ± 0.8 | 20.0 ± 0.9 |
| Height (cm) | 167.4 ± 4.4 | 173.4 ± 5.1[*] |
| Mass (kg) | 64.3 ± 4.0 | 67.6 ± 5.3[*] |
| BMI (kg/m$^2$) | 22.9 ± 0.7 | 22.5 ± 1.0 |
| Table tennis training and competition experience (years) | 4.4 ± 1.2 | 10.0 ± 1.6[**] |

Notes:
BMI, body mass index (calculated as the weight in kilograms divided by the square of the height in meters). Values are given as mean ± SD.
[*] $p < 0.05$.
[**] $p < 0.01$ (student-$t$ test, two-tails).

*Chow, Chen & Wang, 2018*; *Cavanagh & Rodgers, 1987*; *Cornwall & McPoil, 2004*; *Ribeiro et al., 2016*).

## Statistical analysis

The study used descriptive statistical methods to analyze the anthropometric characteristics of eligible participants, and values for the analysis was presented as mean ± standard deviation (SD) and summarized in Table 1. Participants in the two groups were significantly different in their mean height, mass and experiences of table tennis training and competition at the 95% confidence level. The study further conducted an independent sample $t$-test on the participants of the two groups to compare the parameters of AI values and plantar loads distribution in each region and centers of gravity balance as well as the rearfoot postural alignment. Additionally, the paired-samples $t$-test was used to analyze the transitional changes of plantar loads between the static stance and the walking midstance phase in each group. Statistical significance for tests in the present study was represented by $P < 0.05$ (marked with [*]) and $P < 0.01$ (marked with [**]), and was conducted by using the SPSS software (IBM SPSS Statistics 21.0, Armonk, NY, USA).

## RESULTS

### Bipedal arch index

The results of the static arch indices of both feet showed that there were no significant differences between the groups. The participants in each group had normal arch indices and symmetrical feet, suggesting that they belonged to normal arched foot (Table 2).

### Bipedal plantar load distributions of the five regions

Plantar pressure distributions in the present study were calculated as percentages of the relative plantar loads. As a result of static standing, the elite group's plantar loads were predominantly distributed to the bipedal forefoot and right-lateral foot ($p < 0.01$), and relatively lower to the bipedal medial foot compared to the amateur group ($p < 0.05$) (Table 3). Additionally, during the midstance phase of walking, the elite group's plantar loads were primarily exerted on the bipedal forefoot, rearfoot and lateral foot ($p < 0.05$). Based on the comparison of the changes between the static stance and the walking midstance phase in each group, the results showed that the amateur group's plantar loads

**Table 2  Bipedal arch indices of elite and amateur table tennis players.**

|  | Amateur group ($n$ = 77) | Elite group ($n$ = 70) | $p$-value |
|---|---|---|---|
| Left foot | 0.21 ± 0.03 | 0.22 ± 0.07 | 0.121 |
| Right foot | 0.21 ± 0.04 | 0.22 ± 0.07 | 0.103 |

Note:
The static bipedal arch indices of both groups are represented as mean ± SD. Statistical significance of $p$-values was determined by the independent sample $t$-test.

**Table 3  Bipedal plantar load distributions of the five regions under static and dynamic states.**

| Five regions | Static standing | | Midstance phase of walking | |
|---|---|---|---|---|
|  | Left foot (%) | Right foot (%) | Left foot (%) | Right foot (%) |
| **Amateur group ($n$ = 77)** | | | | |
| Forefoot | 25.39 ± 5.37 | 25.69 ± 5.21 | 22.24 ± 4.06[d] | 22.56 ± 3.99[d] |
| Midfoot | 9.27 ± 8.44 | 8.95 ± 8.17 | 10.81 ± 9.94[d] | 10.93 ± 10.06[d] |
| Rearfoot | 15.34 ± 5.31 | 15.36 ± 5.02 | 16.95 ± 5.50[c] | 16.52 ± 4.24[d] |
| Lateral foot | 20.56 ± 5.77 | 20.34 ± 5.77 | 20.97 ± 4.41[d] | 20.96 ± 4.09[d] |
| Medial foot | 12.77 ± 10.69 | 12.99 ± 10.69 | 12.37 ± 9.42[a] | 12.38 ± 9.03[d] |
| **Elite group ($n$ = 70)** | | | | |
| Forefoot | 26.43 ± 4.83[b] | 26.50 ± 6.15[b] | 22.62 ± 4.96[b,d] | 22.81 ± 4.59[a,d] |
| Midfoot | 10.80 ± 10.33 | 10.23 ± 9.72 | 10.02 ± 9.03[d] | 10.02 ± 9.15[c] |
| Rearfoot | 12.77 ± 4.93 | 13.27 ± 5.37 | 17.36 ± 5.25[b,d] | 17.17 ± 5.56[b,d] |
| Lateral foot | 22.01 ± 6.14 | 22.66 ± 7.01[b] | 21.13 ± 4.97[a,d] | 21.11 ± 5.01[a,d] |
| Medial foot | 11.32 ± 10.23[a] | 10.67 ± 9.27[a] | 12.20 ± 8.82 | 12.22 ± 8.96 |

Note:
Bipedal plantar load distributions of the five regions under both states are represented as a percentage of relative plantar load and the values are expressed as mean ± SD. Statistical significances of $p$-values ([a]$p < 0.05$ and [b]$p < 0.01$) between both groups were determined by the independent sample $t$-test, while $p$-values ([c]$p < 0.05$ and [d]$p < 0.01$) between the static stance and the walking midstance phase in each group were determined by the paired-samples $t$-test.

were significantly transferred to the bipedal midfoot, rearfoot and lateral foot ($p < 0.05$), but lower on the forefoot and medial foot ($p < 0.05$). In contrast, the elite group's plantar loads were mainly shifted to the bipedal rearfoot ($p < 0.01$) while the forefoot, midfoot, and lateral foot loads were relatively decreased ($p < 0.05$).

## Bipedal plantar load distributions of the six subregions

The six subregional plantar load distributions were divided from the five regions. For the results of static standing, the elite group's plantar loads were symmetrically distributed to the bipedal lateral metatarsals, and the plantar loads of bipedal medial metatarsals and medial heels were relatively decreased compared to the amateur group ($p < 0.05$) (Table 4). Yet, during the midstance phase of walking, their plantar loads shifted to be applied under the bipedal lateral metatarsals, medial and lateral heels ($p < 0.05$). As for the transitional changes from static standing to the walking midstance phase in each group, the amateur group bore larger plantar loads at the bipedal medial longitudinal arches and the lateral heel of the left foot, but reduced plantar loads at the medial and lateral metatarsals of both feet ($p < 0.05$). For the elite group, however, the plantar loads shifted toward the bipedal

**Table 4 Bipedal plantar load distributions of the six subregions under static and dynamic states.**

| Six subregions | Static standing | | Midstance phase of walking | |
|---|---|---|---|---|
| | Left foot (%) | Right foot (%) | Left foot (%) | Right foot (%) |
| Amateur group ($n$ = 77) | | | | |
| Lateral metatarsal bone (LM) | 25.72 ± 4.47 | 26.08 ± 4.42 | 23.13 ± 3.37[c] | 23.31 ± 3.53[c] |
| Lateral longitudinal arch (LLA) | 17.31 ± 3.51 | 16.63 ± 3.78 | 20.31 ± 3.95 | 20.48 ± 4.28 |
| Lateral heel (LH) | 18.65 ± 3.92 | 18.32 ± 3.95 | 19.47 ± 4.94[c] | 19.07 ± 3.22 |
| Medial metatarsal bone (MM) | 25.06 ± 6.16 | 25.31 ± 5.91 | 22.55 ± 4.65[d] | 21.80 ± 4.29[d] |
| Medial longitudinal arch (MLA) | 1.23 ± 0.52 | 1.27 ± 0.84 | 1.31 ± 0.82[c] | 1.37 ± 0.57[c] |
| Medial heel (MH) | 12.03 ± 4.38 | 12.39 ± 4.17 | 13.23 ± 4.10 | 13.97 ± 3.55 |
| Elite group ($n$ = 70) | | | | |
| Lateral metatarsal bone (LM) | 28.85 ± 3.30[a] | 31.08 ± 3.35[a] | 24.79 ± 4.21[a,c] | 24.50 ± 4.32[b,d] |
| Lateral longitudinal arch (LLA) | 20.82 ± 3.32 | 19.51 ± 3.91 | 18.48 ± 4.37[d] | 18.62 ± 4.26[c] |
| Lateral heel (LH) | 16.35 ± 3.29 | 17.39 ± 3.46 | 20.12 ± 4.02[a,c] | 20.23 ± 4.54[a,c] |
| Medial metatarsal bone (MM) | 24.00 ± 4.92[a] | 21.91 ± 4.73[b] | 20.45 ± 4.72 | 21.11 ± 4.24 |
| Medial longitudinal arch (MLA) | 0.78 ± 0.48 | 0.96 ± 0.58 | 1.57 ± 0.49 | 1.42 ± 0.65 |
| Medial heel (MH) | 9.19 ± 3.48[a] | 9.15 ± 3.41[a] | 14.60 ± 4.88[a,d] | 14.12 ± 4.76[a,d] |

Note:
Bipedal plantar load distributions of the six subregions under both states are represented as a percentage of relative plantar load and the values are expressed as mean ± SD. Statistical significances of $p$-values ([a]$p < 0.05$ and [b]$p < 0.01$) between both groups were determined by the independent sample $t$-test, while $p$-values ([c]$p < 0.05$ and [d]$p < 0.01$) between the static stance and the walking midstance phase in each group were determined by the paired-samples $t$-test.

**Table 5 Static centers of gravity balance under static standing state.**

| | Amateur group ($n$ = 77) | Elite group ($n$ = 70) | $p$-value |
|---|---|---|---|
| Left foot (%) | 50.15 ± 5.46 | 50.03 ± 4.47 | 0.289 |
| Right foot (%) | 49.85 ± 5.46 | 49.97 ± 4.47 | 0.289 |

Note:
The statistics of the percentage of bipedal static centers of gravity are expressed as mean ± SD. Statistical significance of $p$-values was determined by the independent sample $t$-test between both groups.

**Table 6 Static rearfoot postural alignment under static standing state.**

| | Amateur group ($n$ = 77) | Elite group ($n$ = 70) | $p$-value |
|---|---|---|---|
| Left foot (deg.) | 4.04 ± 1.69 | 2.73 ± 2.30 | 0.021 |
| Right foot (deg.) | 4.07 ± 1.68 | 2.70 ± 2.32 | 0.010 |

Note:
The statistics of static angles (°) of bipedal rearfoot postural alignment are represented as mean ± SD and $p$-values were determined by the independent sample $t$-test between both groups.

medial and lateral heels, and were relatively lower at the lateral parts of metatarsals and longitudinal arches ($p < 0.05$).

## Static bipedal centers of gravity balance

The static centers of gravity distributions in the study were calculated as a percentage of gravity and represented the balance situation of the participants' feet. The results showed

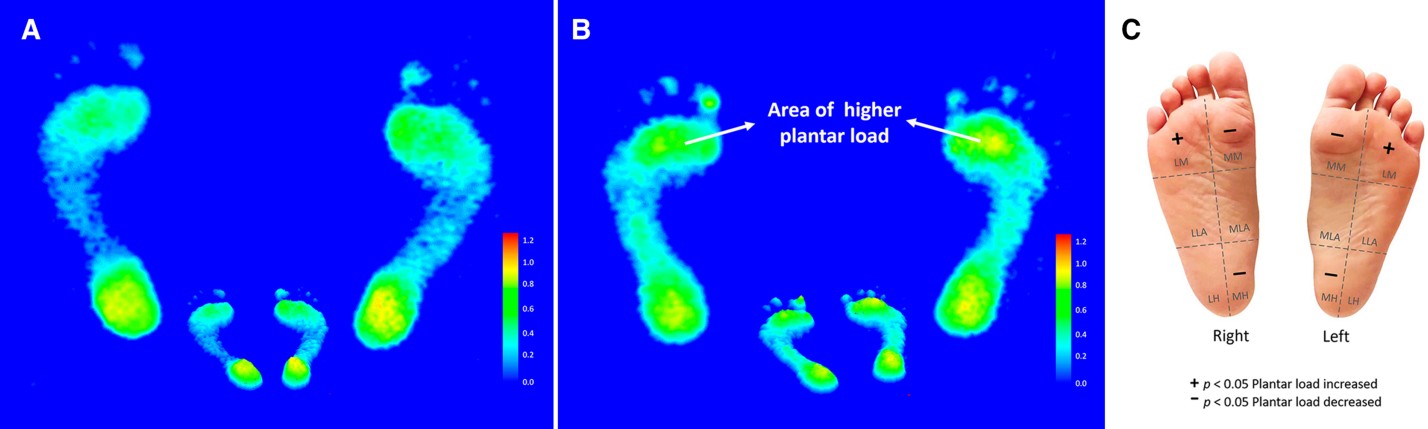

**Figure 1** The color footprint image of the homogenized representative subject was presented by averaging the six subregional plantar loads of participants in the amateur group ($n = 77$) (A). The color footprint image of the homogenized representative subject was presented by averaging the six subregional plantar loads of participants in the amateur group ($n = 77$) (A) and the elite group ($n = 70$) (B). An identifiable plantar diagram illustrates the bipedal plantar load distributions of the elite group (C). The six subregions of the plantar are abbreviated as follows: MM, medial metatarsal bone; MLA, medial longitudinal arch; MH, medial heel; LM, lateral metatarsal bone; LLA, lateral longitudinal arch and LH, lateral heel. A statistically significant increase or decrease in plantar load is indicated by a + or − symbol.     

that the static centers of gravity on both feet of each group were symmetrical and balanced, and there were no significant differences between the groups (Table 5).

## Static bipedal rearfoot postural alignment

The static angles of bipedal rearfoot postural alignment were measured in degrees (deg). The bipedal rearfoot postural angles of each group were symmetrical and within the normal range upon static standing. Additionally, the bipedal average rearfoot angles of the elite group were significantly lower than those of the amateur group ($p < 0.05$) (Table 6).

## Footprint image characteristics

The footprint image of the homogenized representative subject was presented by averaging the plantar load distributions of participants within each respective group. The results showed that the static footprint characteristics of the elite group revealed the higher plantar load mainly distributed on the bipedal forefoot regions, particularly the lateral metatarsals (Fig. 1). Referring to the statistical results, the corresponding identifiable plantar diagram illustrated that the plantar loads of the elite group at the bipedal medial metatarsals and medial heels were significantly lower than that of the amateur group.

## DISCUSSION

Since the vigorous advancement of sports science and the in-depth application of biomechanical measurements in exercise training and competitions have accelerated in recent years, there are many countries worldwide that are paying attention to more sophisticated and accurate sports analyses of various sports. Table tennis is one of the popular and eye-catching sports disciplines. The study is different from other previous research that typically focuses on professional footwork and stroke performance of table tennis. This study aimed to investigate the daily static and dynamic plantar load

distributions and centers of gravity balance performance in Taiwanese college-level elite table tennis players. Through the results, we could further understand the outcome of the influence of years of long-term high-load training and competition experiences on the plantar load distribution performance of these elite athletes.

The main findings of the present study were that the elite group's bipedal static arch indices were mutually symmetric in their daily lives and classified as normal arches. Their relative plantar loads were mainly concentrated on the bipedal forefoot and right-lateral foot when standing statically, and relatively lower to the bipedal medial foot compared to the amateur group. While examining plantar loading during the walking midstance phase at their daily habitual pace, the loads were dispersed evenly among the bipedal forefoot, rearfoot and lateral foot. Furthermore, when in the transition state between static stance and walking midstance phase, the plantar loads of the elite group were concentrated entirely on the bipedal rearfoot regions, while lower on the forefoot, midfoot and lateral foot. The findings of this study seem to echo previous research showing that in basic table tennis footwork, the heels of skilled athletes can provide robust propulsion, thereby protecting the heels by counteracting ground reaction forces (*Shao et al., 2014*).
The stretching of the Achilles tendon caused by long-term heel propulsion could apply the supporting load of the forefoot on the ground, which may lead to chronic fatigue and severe wear of the forefoot of the sole (*Yang, 2013*). Furthermore, the results also partly support the previous research that professional athletes with decreased anterior-posterior plantar pressure excursion during the front-end stage, and increased contact areas on the forefoot, midfoot and rearfoot, while decreased at the hallux region (*Fu et al., 2016*). Although the results were also somewhat similar to the study of *Wong, Lee & Lam (2020)* in that, they found that when professional athletes performed forehand loops during the backward end phase, they showed a greater plantar pressure excursion in the medial-lateral direction than anterior-posterior, which was intended to compromise agility and dynamic stability; however, some differences between these studies and our results were that they showed an increase in the contact area of the midfoot and a decrease in the contact area of the little toe.

In addition, the results of six subregional plantar load distributions indicated that static relative plantar loads of the elite group were symmetrically gathered at the bipedal lateral metatarsals of the forefoot, while during the walking midstance phase, the loads were thus shifted to the bipedal lateral metatarsals of the forefoot as well as the entire heels of the rearfoot. As for the transitional changes between both states, the plantar loads were mainly applied on the medial and lateral heels of the rearfoot. Such a layout was inconsistent with those of *Lam et al. (2019)*. This may be based on differences in footwork, distance and gait posture in the research methods, which lead to differences in momentum and strength of force upon landing. However, the study conducted by *Yu et al. (2019)* was the first to observe that the higher peak pressures in professional table tennis players were primarily on the three regions of the medial and lateral rearfoot and the lateral forefoot. As a result of this plantar load distribution pattern, the athlete's body weight will be more evenly distributed across the sole of the foot. The same result was further confirmed by recent studies that professional table tennis players with higher peak plantar pressure on the

medial and lateral rearfoot as well as the lateral forefoot during the one-step phase (*He et al., 2021*; *Yuqi et al., 2022*). Relevant studies have also supported that professional table tennis players are accustomed to landing on the rearfoot during the phase I footwork movement, which results in the maximum plantar pressure being exerted on the three regions. These findings could suggest that professional athletes are able to distribute their centers of gravity well across their entire sole plane, providing a more stable foundation for the next stage of their stroke performance (*Shao et al., 2020*; *Chi, 2020*). In addition, *Li (2022)* also suggested that the middle of the forefoot is the relatively stable main force area of the forefoot in table tennis players from a biomechanical study of asynchronous foot movement. *He et al. (2021)* further elaborated that the greater peak pressure exerted on the medial and lateral rearfoot as well as the lateral forefoot during the one-step phase may lead to the transfer of the centers of gravity to the dominant leg during landing, and the accompanying transfer of energy may lead to the dominant leg bears more load during the process. *Lam et al. (2019)* mentioned that table tennis players are highly dependent on the strength of the dominant leg, which makes the dominant leg prone to injury due to overuse and excessive plantar pressure.

However, as for the performance of centers of gravity, the results of this study showed that the bipedal centers of gravity of the elite group were symmetrical and evenly balanced when standing in a static posture. The results appear to be contrary to *He et al. (2021)*'s argument that such a layout may result in a shift of the centers of gravity to the dominant leg. *Iordan, Mereuță & Mircea (2020)* also mentioned that junior female table tennis players had a slight asymmetry in their shoulders and scapula resulting in a musculoarticular postural imbalance due to specific unilateral movements. *Li (2022)* went further and noted that as the training time is extended for multiball table tennis practice, the dynamic balance ability of the body will gradually decline and thus affecting the athlete's exercise performance. For the situation, however, female athletes' balance skills will decline earlier than male athletes (*Li, 2022*). Although these arguments were inconsistent with our results, our findings also reflect the fact that professional table tennis players possess smooth chasse and stable centers of gravity transfer (*Yu et al., 2019*), and professionally mastered the abilities of predominant foot and body balance control as well as technical stereotypes (*Cetin, 2020*). *Li (2022)* also agreed that professional table tennis players have more stable weight transfer ability in parallel gait. Moreover, *Yu et al. (2019)* realized that professional table tennis players have better foot and body balance control capabilities to smoothly connect the next stage of the kinetic chain, those mainly based on showing less rearfoot varus and forefoot valgus as well as greater hallux plantarflexion.

Based on these associations, we examined the results of the rearfoot posture alignment assessment. The study showed that the bipedal average rearfoot angles of the elite group were symmetrical and within the normal range, that is, the subtalar joint was at the neutral position, but the angle analysis was still statistically lower than that of the amateur group. A study by *Qian et al. (2016)* suggests that the difference in intrinsic joint motion behaviors of the lower extremity may affect the speed of the table tennis racket during the forehand loop of table tennis for advanced and intermediate athletes. To a certain extent, the phenomenon coincides with *Li (2022)*'s arguments that professional athletes with greater

internal rotation of the rearfoot at the end of the lead accompanied by greater abduction of the forefoot are beneficial for athletes to better complete their gait and maintain stable centers of gravity balance.

As a result of the comparison between elite athletes and amateur athletes, the current study may serve as a research theoretical basis for understanding the unique foot pattern characteristics in Taiwanese college elite table tennis players. The preliminary results not only initiate a causal relationship between changes in plantar loads and lower limb inertial movement behaviors among elite athletes, but also outline the potential changes in plantar loads that amateurs may experience the course when they become elites through long-term training and competition. The present study was limited by the fact that it only sampled the plantar loading analysis of 70 elite males and 77 same-gender amateur table tennis players aged 19 to 21 years from Taiwanese colleges and universities. Since the research object of this study was based on a non-random sample of participants, participants selected in the study do not fully represent all elite table tennis players, so the existence of sampling bias may affect the interpretation of the results. In addition, this cross-sectional study was conducted in the classrooms of each participant's school during their non-training and non-competition periods. Therefore, it is inevitable that there may be internal or external influencing factors that are different from the actual practice and competition environment. Furthermore, the study failed to catch a consideration point that only the dominant hand of the participants was considered during the study, but it did not record whether the participants had dominant/non-dominant legs. Therefore, further research suggests that the application of electromyography (EMG) can be considered to determine the static and dynamic signal expression of the dominant leg at their daily habitual paces, and even explore the correlation among plantar load distribution, centers of gravity distribution and lower limb strength of athletes in trained professional footwork.

## CONCLUSIONS

The bipedal static arch indices of the Taiwanese college elite table tennis players in the present study were symmetric and could be classified as normal arches during their non-training and non-competition daily lives. Their static plantar loads were symmetrically gathered at the bipedal lateral metatarsals, while the load shifted and distributed at the three regions of the medial and lateral heels as well as the lateral metatarsals during the walking midstance phase. The plantar loads were transferred to the bipedal medial and lateral heels during the transitional changes between both states. Additionally, the centers of gravity of elite athletes were symmetrical and evenly distributed over their feet when standing in a static posture. The bipedal average rearfoot angles were also symmetrically within the normal range even though it's statistically lower than that of the amateur group. The results suggest that plantar loading, centers of gravity balance and rearfoot posture in the daily lives of elite athletes did not become unilaterally loaded as a result of long-term unilateral movement behavior. This study not only provides traceable information on the changes in plantar pressure distribution experienced by amateur players as they become elite athletes through repeated training or competition, but also reveals the difference in daily foot patterns between elites and ordinary people.

## ACKNOWLEDGEMENTS

We sincerely appreciated to all participants from National Taiwan Normal University, National Taiwan University of Sport, University of Taipei, National United University, National University of Tainan, National Taitung University, National Formosa University, National Taipei University of Technology, National Kaohsiung University of Science and Technology, National Pingtung University of Science and Technology, Fu Jen Catholic University, Providence University, Chinese Culture University, Feng Chia University, Shih Chien University and St. John's University for their contributions to the study.

### Funding

This work was supported by the Ministry of Science and Technology, Taipei, Taiwan under Grant MOST 104-2622-H-129-001-CC3. The funders had no role in study design, data collection and analysis, decision to publish, or preparation of the manuscript.

### Grant Disclosures

The following grant information was disclosed by the authors:
Ministry of Science and Technology, Taipei, Taiwan: MOST 104-2622-H-129-001-CC3.

### Competing Interests

The authors declare that they have no competing interests.

### Author Contributions

- Tong-Hsien Chow conceived and designed the experiments, performed the experiments, analyzed the data, prepared figures and/or tables, authored or reviewed drafts of the article, and approved the final draft.
- Yu-Ling Lee performed the experiments, authored or reviewed drafts of the article, and approved the final draft.

### Human Ethics

The following information was supplied relating to ethical approvals (*i.e.*, approving body and any reference numbers):

This research ethics was approved by National Taiwan University's research ethics committee (IRB number: 201506ES016).

### Data Availability

The raw measurements are available in the Supplemental File.

### Supplemental Information

Supplemental information for this article can be found online at http://dx.doi.org/10.7717/peerj.17173#supplemental-information.

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
