# Peer review of "Plantar load distribution with centers of gravity balance and rearfoot posture in daily lives of Taiwanese college elite table tennis players: a cross-sectional study"

_PeerJ, doi:10.7717/peerj.17173_

## Round 0.1 · original submission · Major Revisions

Dear authors,

Taking into account the comments made by the reviewers of your manuscript, we inform you that it may be accepted, however, it will require an in-depth review, in accordance with the suggestions mentioned.

Best regards

Reviewer 1 ·

Basic reporting

Abstract:
1.Please add the basic physiological information for study participants. 2.Please explain why natural gait walking was chosen for this study rather than other table tennis-specific footwork.

Experimental design

Descriptions of inclusion and exclusion criteria are adequate. Given the study design, I suggest that sampling bias needs to be included in the limitations section.

Validity of the findings

The results are clear and well presented with appropriate statistical analysis. The conclusion is conclusive. Is there any practical implication besides summarizing?

Annotated reviews are not available for download in order to protect the identity of reviewers who chose to remain anonymous.

Reviewer 2 ·

Basic reporting

The papers present novel data regarding plantar load distribution in elite table tennis players. Mainly the methods section’ detail can be improved. Some specific comments bellow.

Abstract
- Present on the results some objective data/values.

Introduction
- L107 – include “years old”

Material and methods
L190 – present the dynamic procedure for plantar load distribution analysis. What moment of the midstance phase was considered, and how the authors confirm repeatability of the measure.

- L197 – write in the past
- L199 – write plantar loads distribution
- What was the sample frequency of the system Biomech 2019…
- The variables in study should be described.
- What is the data collected on the static and on the dynamic analysis?
- Present the rationale to analyse data base on 5 regions and 6 subregions. Also how the 6 subregions derived from the 5 regions?
- The statistical analysis do not mention how comparison between static and dynamic analysis was performed.

Results
- L234 – How it was calculated.
- L246 – The material and methods section do not present how the 6 subregions derived from the 5 regions.

Table 3
Title mentions data from 5 regions but the table mentions 6 subregions. Why the sum of the percentages is not 100?

Experimental design

Methods section gain with more detail.

Validity of the findings

In accordance.

---

## Round 0.2 · accepted · Accept

Dear authors,

The paper has been revised according to the reviewers' suggestions and can be accepted in its current form.

Best regards

Reviewer 1 ·

Basic reporting

The authors have revised the article according to the review comments and suggest that it should be published.

Experimental design

The authors have revised the article according to the review comments

Validity of the findings

no comment

Additional comments

no comment

Reviewer 2 ·

Basic reporting

The paper has improved due to reviewers comments. My suggestion is for paper acceptance..

Experimental design

The text was improved.

Validity of the findings

In accordance.